# Aptamer-Modified Erythrocyte Membrane-Coated pH-Sensitive Nanoparticles for c-Met-Targeted Therapy of Glioblastoma Multiforme

**DOI:** 10.3390/membranes12080744

**Published:** 2022-07-29

**Authors:** Xianping Liu, Yixin Chen, Daoying Geng, Haichun Li, Ting Jiang, Zimiao Luo, Jianhong Wang, Zhiqing Pang, Jun Zhang

**Affiliations:** 1Department of Radiology, Huashan Hospital, State Key Laboratory of Medical Neurobiology, Fudan University, 12 Wulumuqi Middle Road, Shanghai 200040, China; liuxianping0109@163.com (X.L.); chenyixin0230@163.com (Y.C.); daoyinggeng@163.com (D.G.); 2National Center for Neurological Disorders, 12 Wulumuqi Middle Road, Shanghai 200040, China; 3School of Pharmacy, Fudan University, Key Laboratory of Smart Drug Delivery, Ministry of Education, 826 Zhangheng Road, Shanghai 201203, China; haichun.li@stemirna.com (H.L.); 13554460571@163.com (T.J.); luo_zimiao@163.com (Z.L.); 4Department of Neurology, Huashan Hospital, State Key Laboratory of Medical Neurobiology, Fudan University, 12 Wulumuqi Middle Road, Shanghai 200040, China

**Keywords:** glioblastoma multiforme (GBM), aptamer, c-Met, biomimetic drug delivery system, chemotherapy

## Abstract

Biomimetic drug delivery systems, especially red blood cell (RBC) membrane-based nanoparticle drug delivery systems (RNP), have been extensively utilized in tumor drug delivery because of their excellent biocompatibility and prolonged circulation. In this study, we developed an active targeting pH-sensitive RNP loaded with DOX by decorating an aptamer SL1 on RBC membranes (SL1-RNP-DOX) for c-Met-targeted therapy of glioblastoma multiforme (GBM). SL1 could specifically bind to c-Met, which is highly expressed in GBM U87MG cells and facilitate DOX delivery to GBM cells. In vitro studies demonstrated that U87MG cells had a higher uptake of SL1-RNP-DOX (3.25 folds) and a stronger pro-apoptosis effect than unmodified RNP-DOX. In vivo fluorescence imaging and tissue distribution further demonstrated the higher tumor distribution of SL1-RNP-DOX (2.17 folds) compared with RNP-DOX. As a result, SL1-RNP-DOX presented the best anti-GBM effect with a prolonged median survival time (23 days vs. 15.5 days) and the strongest tumor cell apoptosis in vivo among all groups. In conclusion, SL1-RNP-DOX exhibited a promising targeting delivery strategy for GBM therapy.

## 1. Introduction

Glioblastoma multiform (GBM) is the most common primary malignant tumor of the central nervous system, with high incidence, high recurrence, high mortality, and low cure rate. In recent years, although the sequential standardized treatment for patients initially diagnosed with GBM, surgical resection combined with postoperative adjuvant chemotherapy and radiotherapy has made progress, the prognosis remains poor, and the average survival time is only 12–15 months [1,2,3]. The invasive GBM grows rapidly and infiltrates into the normal brain with the uncertain edge of tumors, which makes surgical resection impossible to remove all parenchymal portions of the malignant GBM, and the tumor recurrence rate is high [4]. Moreover, due to the heterogeneity of GBM and the existence of the blood-brain barrier (BBB), which prevents chemotherapeutics delivery into the glioma tissue, chemotherapeutic drugs are ineffective and prone to inducing drug resistance. Therefore, the anti-GBM treatment remains a great challenge and elusive [5].

To solve this problem, nanoparticulate drug delivery systems have attracted increasing attention in recent years [6]. With the development of nanomaterials, many nanomaterials such as liposomes, polymeric nanoparticles, solid lipid nanoparticles, and polymeric micelles have been extensively explored to cross the BBB [7,8]. In our previous study, a multimodal imaging nanoprobe, PEPHC1-modified superparamagnetic iron oxide nanoparticles, was constructed to diagnose GBM [9]. Like most convention nanoparticles, this nanoprobe has limitations, such as short circulation time and low brain tumor targeting. Recently, biomimetic drug delivery systems, especially red blood cell (RBC) membrane-based nanoparticle drug delivery systems (RNP), have been extensively utilized in tumor drug delivery because of their excellent biocompatibility and prolonged circulation [10,11,12]. Thanks to preserving the complete “self-makers” (e.g., CD47 proteins, acidic sialyl moieties, glycans) of erythrocyte membranes on RNP, RNP can escape immune cell recognition and uptake, resulting in a longer circulation time than PEGylated nanoparticles and no immunological responses [13,14]. RNP, usually consisting of a biodegradable and biocompatible poly (lactic-co-glycolic acid) (PLGA) nanoparticle core and an RBC membrane shell, represents a promising brain drug delivery system due to their combination of high drug carrying capacity along with an inherently biocompatible membrane coating.

c-Met (proto-oncogene protein) is a prototypic member that belongs to a family of the receptor tyrosine kinase (RTKs), and the nature ligand for c-Met is hepatocyte growth factor/scatter factor (HGF/SF) produced mainly by mesenchymal cells [15,16]. In addition, c-Met deregulation has been identified in cancer biology of many human cancer types. c-Met is mainly overexpressed on GBM of the mesenchymal phenotype. It can affect tumorigenicity and is closely associated with the poor clinical prognosis of GBM. In addition, c-Met plays a significant role in mediating radiation resistance and tumor progression in GBM and has been suggested as a prime target for GBM therapy [17,18]. 

Aptamers, first reported in the 1990s, are short RNA or single-stranded DNA oligonucleotides [19,20], which have specificity and high affinity to bind to their targets by recognizing the unique secondary or three-dimensional structures [21,22,23]. Compared with peptides or antibodies, aptamers possess particular advantages, including convenient synthesis, smaller size, easy modification, good stability, and non-immunogenicity. They have been performed as targeting ligands for diagnosis and therapeutics [24]. CLN0003_SL1 (SL1), a 50-mer DNA aptamer that specially binds to c-Met protein, is found to inhibit HGF-induced c-Met activation [25] and may efficiently mediate nanoparticle delivery into intracranial GBM.

Here, we designed an active targeting pH-sensitive RNP loaded with DOX by decorating an aptamer SL1 on RBC membranes (SL1-RNP-DOX) for c-Met-targeted therapy of GBM (Figure 1). DOX was encapsulated into PLGA nanoparticles via a non-surfactant emulsion method [26] under a basic environment and then coated with SL1-modified RBC membranes. Due to the reprotonation of PLGA-COOH in the acid lysosomes, PLGA nanoparticle cores could dissemble to rapidly release DOX and demonstrate a pH-sensitive behavior. In vitro cellular uptake, viability test, and apoptosis assay were performed to identify the targeting effect of SL1-RNP-DOX. In vivo imaging and tissue distribution were utilized to evaluate the GBM targeting effect of SL1-RNP-DOX. To further demonstrate the chemotherapy value of SL1-RNP-DOX, a survival study of the brain glioma-bearing mice was performed.

## 2. Materials and Methods

### 2.1. Materials

Carboxylic acid-terminated PLGA (PLGA-COOH; 0.67 dL/g, 50:50 ratio) was purchased from Lactel (Cupertino, CA, USA). Doxorubicin hydrochloride (DOX∙HCL) was ordered from Beijing Huafeng United Technology (Beijing, China). The 5′-NH_2_-C6-SL1 DNA aptamer (SL1 squence: 5′—3′, ATCAGGCTGGATGGTAGCTCGGTCGGGGTGGGTGGGTTGGCAAGTCTGAT 50 bp) was synthesized by Sangon Biochemistry Company (Shanghai, China). DNase/RNAase-free water was purchased from Rainbio (Shanghai, China). Poly(oxy-1,2-ethanediyl) succinimidyl ester, α-(carboxymethyl)-ω-[[(10R)-7-hydroxy-7-oxido-2,13-dioxo-10-[(1-oxooctadecyl)oxy]-6,8,12-trioxa-3-aza-7-phosphatriacont-1-yl]oxy] (DSPE-PEG_2000_-NHS) was purchased from Xi’an Ruixi Biological Technology (Xi’an, China).Quant-iT PicoGreen dsDNA Assay Kit was purchased from Invitrogen (USA). 3-(4,5-Dimethylthiazol-2-yl)-2,5-diphenyltetrazdium bromide (MTT) Cell Proliferation and Cytotoxicity Assay Kit was purchased from Sigma (St. Louis, MO, USA). 1,10-dioctadecyl-3,3,30,30-tetramethylindo-tricarbocyanine iodide (DiR) was obtained from AAT Bioquest (Sunnyvale, CA, USA). Coumarin 6 was supplied by Sigma (USA). 4,6-diamidino-2-phenylin dole dihydrochloride (DAPI) was purchased from Beyotime (Nantong, China). The terminal deoxynucleotide transferase-mediated dUTP nick end-labeling (TUNEL) detective kit was purchased from Roche (Solna, Sweden). The Annexin V-FITC apoptosis detection kit was purchased from Dojindo Molecular Technologies (Kumamoto, Japan). An ultracentrifuge filter device with a molecular weight cutoff (MWCO) of 30/50 kDa was purchased from Millipore (Bedford, MA, USA). Dulbecco’s Modified Eagle’s Medium (high glucose) (DMEM, Gibco), fetal bovine serum (FBS), trypsin-EDTA (0.25%), and penicillinestreptomycin were purchased from Gibco (CA). Purified deionized water (Millipore, Bedford, MA, USA) was used throughout the study. Plastic cell culture dishes and plates were obtained from Corning Incorporation (Corning, NY, USA). All other reagents and chemicals were of analytical reagent grade and purchased from Sinopharm Chemical Reagent (Shanghai, China).

### 2.2. Cells and Animals

The U87MG cell line and human umbilical vein endothelial cells (HUVECs) were obtained from the Institute of Biochemistry and Cell Biology, Shanghai Institutes for Biological Sciences, Chinese Academy of Sciences (Shanghai, China). The cells were cultured at 37 °C, 5% CO_2_ in special DMEM supplemented with 10% FBS, 100 IU/mL penicillin, and 100 mg/mL streptomycin.

Male BALB/c nude mice (4–5 weeks, 18–20 g) and BALB/c mice (25–30 g) were purchased from Shanghai Slac Laboratory Animal Ltd. (Shanghai, China) and raised under a standard condition (25 ± 2 °C and 60% ± 10% humidity enviroment with 12-h light/dark cycle). All animal experiments were performed in accordance with protocols evaluated and approved by the Animal Ethics Committee of Fudan University (2017-03-YJ-PZQ-01). The intracranial glioblastoma animal model was established according to a previously reported procedure [27]. Briefly, 5 × 10^5^ U87MG cells were inoculated into the right striatum using a stereotactic fixation device (Stoteling, Wood Dale, IL, USA) with the coordinates as follows: 2 mm right lateral to the bregma and 4 mm depth from the dura.

### 2.3. Preparation of Nanoparticle Cores

PLGA nanoparticle cores loaded with DOX (NP-DOX) were prepared through a non-surfactant double emulsion method as previously described [26]. Briefly, 10 mg of PLGA-COOH were dissolved in 1 mL of dichloromethane as the organic phage. DOX dissolved in 50 μL of 500 mmol/L Tris-HCl (pH 8.0) was added into the polymer solution and sonicated with a probe sonicator to form a W/O emulsion. Afterwards, the W/O emulsion was added with 5 mL of 10 mmol/L Tris-HCl (pH 8.0) and sonicated again to generate the W/O/W emulsion. The final emulsion was then dispersed in 10 mL of 10 mmol/L Tris-HCl (pH 8.0), and dichloromethane was removed under a ZX-98 rotary evaporator (Shanghai Institute of Organic Chemistry, Shanghai, China) to obtain NP-DOX. Coumarin 6- or DiR-labelled NP was developed with the same procedure, except that 50 µg of Coumarin 6 or 50 µg of DiR was dissolved in 1 mL of the polymer solution in advance.

### 2.4. Preparation of SL1-Modified RBC Membranes

RBC membranes were prepared as previously described [12,28]. Briefly, the fresh whole blood (~2 mL) was collected from BALB/c mice (male, 25–30 g), suspended in 10 mL of heparin-containing phosphate saline buffer solution (PBS, 0.01 mol/L), and centrifuged at 800× *g* for 5 min at 4 °C to remove the serum and the buffy coat. Afterwards, RBCs were washed with ice-cold PBS containing 1 mmol/L EDTA∙2Na three times and recollected by centrifugation 800× *g* at 4 °C. The resulting RBCs (0.25 mL) were then subjected to 1 mL of the hypotonic solution (0.25 mmol/L EDTA∙2Na) for hemolysis and centrifugation at 20,000× *g* three times to remove the hemoglobin. The resulting RBC membranes were collected and suspended in a 0.25 mL hypotonic solution. The membrane protein concentration was quantitated by the bicinchoninic acid method, and around 2.0 mg of membrane proteins per milliliter was obtained in RBC membrane suspensions, which agreed with our previous report [28].

For the synthesis of SL1-modified DSPE-PEG (DSPE-PEG-SL1), DSPE-PEG-NHS was incubated with 5′-NH_2_ SL1 aptamer (1 mg/mL) with an equimolar ratio in DNase/RNAase-free water overnight. SL1-modified RBC membranes (SL1-RBC) were prepared by a lipid insertion method [29]. Briefly, DSPE-PEG-SL1 was incubated with RBC membranes for 30 min at 37 °C to form SL1-RBC. The resulting SL1-RBC was then washed by pelleting at 800× *g* for 5 min at 4 °C before further use. The SL1 density on SL1-RBC was calculated by quantifying the free SL1 in the supernatant with the Quant-iT PicoGreen dsDNA Assay Kit.

### 2.5. Preparation and Characterization of SL1-RNP-DOX

SL1-RNP-DOX was prepared with a sonication method [30]. In brief, SL1-RBC (125 µL) was added to 100 µL of NP-DOX (1 mg) and sonication for 2 min in an ultrasonic water bath. Accordingly, Coumarin 6- or DiR-labelled SL1-RNP was prepared with the same method. Transmission electron microscopy (TEM) (Hitachi, Tokyo, Japan) was used to observe the morphology of SL1-RNP-DOX. Briefly, a drop of SL1-RNP-DOX solution (1 mg/mL) was deposited onto a glow-discharged carbon-coated TEM grid, followed by negative staining with 1% uranyl acetate. The particle size and zeta potential of RNP-DOX and SL1-RNP-DOX were determined using a Malvern Nano ZS (Malvern Instruments, Malvern, UK). The stability of RNP-DOX and SL1-RNP-DOX in vitro was assessed by monitoring the particle size every day during one-week storage at 4 °C in 0.01 mol/L PBS [28]. The leakage of DOX from SL1-RNP-DOX after storage was determined by measuring its fluorescence (excitation at 480 nm; emission at 580 nm) in the supernatant after centrifugation at 20,000× *g* for 15 min.

### 2.6. Drug Loading and Drug Release

The amount of DOX loading in nanoparticles was evaluated by measuring its fluorescence (excitation at 480 nm; emission at 580 nm). The drug loading capacity (DLC) and encapsulation efficiency (EE) of DOX in nanoparticles were calculated as follows:DLC = DOX_encapsulated_/total materials × 100%;
EE = DOX_encapsulated_/DOX_input_ × 100%

Drug release was studied by dialyzing samples in 0.01 mol/L PBS (pH = 7.4 or 5.0) using dialysis tubes (Thermo Scientific, Waltham, MA, USA) with an MWCO of 10 kDa [26].

### 2.7. In Vitro Cytotoxicity Assay

The cytotoxicity of different nanoparticles (NP, RNP, SL1-RNP) against U87MG cells was evaluated by the MTT assay. Briefly, U87MG cells in the logarithmic growth phase were seeded into 96-well plates (2 × 10^3^ cells per well). After 24 h of incubation, cells were incubated with 200 µL of fresh medium containing different nanoparticles with various concentrations (0, 25, 50,100 or 200 µg/mL). To evaluate the cytotoxicity of Free Dox, RNP-DOX, and SL1-RNP-DOX, U87MG cells or HUVECs were seeded and cultured for 24 h as described above. Afterwards, cells were incubated with 200 µL of fresh medium containing Free DOX, RNP-DOX, or SL1-RNP-DOX with the DOX concentration ranging from 1 nmol/L to 0.1 mmol/L. Untreated cells were used as control. After 24-h incubation, the culture medium was discarded, and 180 µL of fresh medium and 20 µL of MTT solution were added to each well. The plate was incubated for an additional 4 h, then the culture medium was discarded, and 150 µL of dimethyl sulfoxide (DMSO) solution was added to each well. The absorbance values were measured under a microplate reader (Bio-TEK, USA) at the wavelength of 490 nm. In addition, the half-maximal inhibitory concentrations (IC50) of different DOX formulations (Free DOX, RNP-DOX, and SL1-RNP-DOX) against U87MG cells or HUVECs were analyzed by GraphPad Prism v6.02.

### 2.8. Cell Apoptosis Assay

The cell apoptosis mediated by RNP-DOX and SL1-RNP-DOX was assessed by flow cytometry. In brief, cells were seeded into 12 well plates at a density of 10^5^ per well and cultured for 24 h. After that, cells were incubated with a fresh medium containing RNP-DOX and SL1-RNP-DOX (0.1 mmol/L DOX) for 24 h and then stained using the Annexin V-FITC apoptosis detection kit. The percent of early apoptosis and late apoptosis were analyzed by flow cytometry (BD, New York, NJ, USA).

### 2.9. In Vitro Cellular Uptake

Cellular uptake of Coumarin 6-labelled RNP or SL1-RNP by U87MG cells and HUVECs were investigated. Briefly, U87MG or HUVECs in the logarithmic phase were seeded at a density of 10^4^ cells per well into 24-well plates. When cells reached 80% confluence, they were incubated with 50 µg/mL Coumarin 6-labelled RNP and SL1-RNP for 60 min, respectively. For qualitative analysis of the cellular uptake, U87MG or HUVECs were washed with 0.01 mol/L PBS three times, fixed with 4% paraformaldehyde for 15 min, and treated with DAPI for 5 min, and then observed under a fluorescent microscope (Leica, DMI 4000B, Wetzlar, Germany). For quantitative analysis, U87MG and HUVECs were collected by trypsin digestion after incubation, suspended in 0.01 mol/L PBS, and subjected to flow cytometry (BD, New York, NJ, USA).

### 2.10. In Vivo Fluorescence Imaging and Biodistribution

GBM-bearing mice were intravenously injected with 200 µL of DiR-labelled RNP and SL1-RNP (100 μg/mL DiR) ten days post glioblastoma implantation. The fluorescence signal acquisitions were performed at various time points (2, 8, 24, and 48 h) after i.v. injection using the In Vivo IVIS spectrum Imaging System (PerkinElmer, Waltham, MA, USA). At forty-eight hours after injection, mice were sacrificed, followed by heart perfusion with saline and the brain and other major organs (hearts, livers, spleens, lungs and kidneys) were collected and ex vivo imaging of these organs was also captured.

Biodistribution of RNP and SL1-RNP in GBM-bearing mice was investigated to assess the targeting property of RNP and SL1-RNP in vivo. Ten days after GBM implantation, mouse models were injected with 200 µL of DiR-labelled RNP and SL1-RNP at a DiR dose of 10 µg via the tail vein, respectively. Twenty-four or forty-eight hours later, mice were sacrificed, followed by heart perfusion with saline. Major organs were harvested, homogenized with saline, and the fluorescence intensity of samples was determined with a Tecan Infinite M200 Pro Multiplate Reader (Switzerland, Ex/Em = 740 nm/780 nm). The concentration of nanoparticles in tissues was expressed as the percentage of injected dose per gram of tissue (% ID/g).

The distribution of RNP and SL1-RNP in GBM slices was investigated to evaluate the distribution pattern of RNP and SL1-RNP in vivo. Ten days after GBM implantation, mouse models were injected with 200 µL of Coumarin 6-labelled RNP and SL1-RNP at a Coumarin 6 dose of 10 µg via the tail vein, respectively. Twenty-four hours later, mice were sacrificed, followed by heart perfusion with saline, and the brains were harvested, dehydrated in 15% and 30% sucrose successively, embedded in OTC, and cut into 10-µm sections. Finally, the sections were stained with DAPI and fluorescence images were captured under a fluorescent microscope (Leica, DMI 4000B, Germany).

### 2.11. Anti-GBM Efficacy in Vivo

Intracranial glioblastoma-bearing nude mice were established as described above. Mice were randomly divided into three groups to receive different treatments, including RNP-DOX, SL1-RNP-DOX (5 mg/kg of DOX), and saline via i.v. injection at 3, 6, 9, and 12 days after U87MG inoculation, respectively. The body weight of tumor-bearing mice was monitored every two days. The survival data were analyzed with the log-rank test in a Kaplan-Meier nonparametric analysis and summarized descriptively using median survival by GraphPad Prism v6.02. On the 12th day, three mice from each group were sacrificed. The brains were harvested, fixed with 4% paraformaldehyde for 48 h, embedded in paraffin, and cut into 10-µm sections. The sections were then subjected to TUNEL staining according to the protocol and visualized under a fluorescent microscope (Leica, Germany).

Major organs, including hearts, livers, spleens, lungs, and kidneys, were obtained from mouse models at the study endpoint and sectioned for H&E staining to preliminarily evaluate the safety of SL1-RNP-DOX. The H&E staining slices were observed under the fluorescence microscope (LEICA DMI4000B, Germany).

### 2.12. Statistical Analysis

GraphPad prism 7 software was used for statistical analysis. Unpaired Student’s *t*-test for two groups’ comparison and one-way analysis of variance (ANOVA) for multiple-group comparison was used to determine differences between groups. Data are presented as mean ± standard deviation, and *p* values < 0.05 were considered statistically different.

## 3. Results

### 3.1. Characterization of SL1-RNP-DOX

The surface morphology of SL1-RNP-DOX was observed under TEM. The nanoparticle core was generally spherical with a regular shape, and the RBC membrane coating could be observed clearly (Figure 1A). The average diameter of RNP-DOX was 85.4 ± 5.6 nm, and SL1 modification on RNP-DOX did not significantly change the size, polydispersity index, and zeta potential (Figure 1B–D). The aptamer SL1 density on RBC membranes was around 4.13 × 10^5^ per RBC. The calculated aptamer SL1 density on SL1-RNP-DOX was 110.5 per nanoparticle, which is similar to the spike density of most viruses [31]. Compared with unstable NP-DOX, SL1-RNP-DOX and RNP-DOX kept stable in PBS at 4 °C and showed no obvious size change during storage (Figure 1E) and less than 5% DOX leakage from SL1-RNP-DOX and RNP-DOX (Appendix A). The drug loading capacity of DOX in SL1-RNP-DOX was around 1.75%, with an encapsulation efficiency of 87.7%. SL1-RNP-DOX demonstrated slow DOX release over time under pH 7.4 (Figure 1F). However, it rapidly released DOX under pH 5.0, and approximately 92% of DOX was released in 24 h, showing a superior pH-sensitive release.

### 3.2. Cytotoxicity Assay and Cell Apoptosis Assay

MTT assay revealed that the viability of U87MG cells after SL1-RNP or RNP treatment was above 80%, even at a high concentration of 200 μg/mL, suggesting both SL1-RNP and RNP have low toxicity on cells (Figure 2A). SL1-RNP-DOX showed increased cytotoxicity on U87MG cells compared with RNP-DOX. The IC50 for free DOX, SL1-RNP-DOX, and RNP-DOX was 0.837, 4.14 and 10.5 µmol/L, respectively (Figure 2B). However, SL1-RNP-DOX and RNP-DOX displayed similar low toxicity on HUVECs (Appendix A), and their IC50 was 117.2 and 128.4 µmol/L, respectively. The pro-apoptosis efficiency of different DOX formulations against U87MG cells was analyzed by flow cytometry (Figure 2C). It was found that SL1-RNP-DOX exhibited a stronger pro-apoptosis activity (23.1%) in U87MG cells compared with that of the RNP-DOX (13.7%) (Figure 2D). 

### 3.3. Cellular Uptake Assay

To investigate the targeting effect of SL1, Coumarin 6 was incorporated into nanoparticles as a sensitive and accurate fluorescence indicator to track nanoparticles in vitro and in vivo [32,33], and fluorescent microscope and flow cytometry were employed to assess the cellular uptake. As shown in Figure 3A, fluorescence images demonstrated that SL1-RNP exhibited significantly higher cellular uptake by U87MG cells than RNP after incubation for 1 h. Flow cytometry showed consistent results with the qualitative analyses (Figure 3B,C). The mean fluorescent intensity of cells for the SL1-RNP group was 3.25 folds higher than that for the RNP group. These results indicated that the aptamer SL1 modification on RNP could significantly enhance the uptake of nanoparticles by U87MG cells. However, it was found that SL1-RNP exhibited cellular uptake similar to RNP by HUVECs after incubation for 1 h (Figure 4A–C), indicating the aptamer SL1 modification on RNP could not enhance the uptake of nanoparticles by HUVECs. Compared with RNP-DOX, the enhanced cytotoxicity and pro-apoptosis activity of SL1-RNP-DOX on U87MG cells could be attributed to the improved cellular uptake of SL1-RNP-DOX by U87MG cells.

### 3.4. In Vivo Fluorescence Imaging and Biodistribution

The accumulation of DiR-labelled RNP and SL1-RNP in orthotopic GBM mouse models was determined by fluorescence imaging. Compared with the RNP group, the fluorescence intensity in the brain of the SL1-RNP group was significantly higher at any time post administration (Figure 5A). Ex vivo fluorescence imaging (Figure 5B–D) demonstrated that the fluorescence signal in the tumor site from the SL1-RNP group was 2.17-fold stronger than that from the RNP group at 48 h, indicating that the SL1 modification greatly enhances the accumulation of SL1-RNP in GBM. Similar to other active targeting nanoparticulate drug delivery systems, RNP and SL1-RNP are mainly distributed in the mononuclear phagocyte system (MPS)-related organs like the liver and the spleen (Figure 5B).

The tissue bio-distribution of SL1-RNP at different time points (24 and 48 h post-injection) was evaluated in BALB/c mice (Figure 6A). Both RNP and SL1-RNP were distributed in organs following the order: liver > spleen > kidney > heart > lung > brain, which agreed with the ex vivo fluorescence imaging results. Like many other nanoparticles, RNP and SL1-RNP were mainly accumulated in the mononuclear phagocyte system (MPS)-related organs, such as the liver and the spleen. There was no significant difference in the biodistribution between SL1-RNP and RNP except for the brain. The accumulation of SL1-RNP in the brain was significantly higher than RNP, probably due to the SL1 modification greatly enhancing the accumulation of SL1-RNP in GBM. The fluorescence images of GBM tissue slices treated with Coumarin 6-labelled RNP or SL1-RNP 24 h post i.v. injection are shown in Figure 6B. It was found enriched SL1-RNP penetrated deeply into the whole GBM tissues with higher fluorescence signals than RNP, suggesting the specific targeting property of SL1-RNP in glioblastoma tissues.

### 3.5. In Vivo Anti-GBM Efficacy

The anti-GBM efficacy of different DOX formulations was investigated by monitoring the survival of intracranial U87MG glioblastoma-bearing nude mice. As shown in Figure 7A, the median survival time of mice in the SL1-RNP-DOX group (23 days) was significantly longer than those in the RNP-DOX group (15.5 days) and the saline group (13 days). The increase in survival time (IST) of the SL1-RNP-DOX group (176.9%) was higher than that of the RNP-DOX group (119.2%) as compared with the saline group. SL1-RNP-DOX notably prolonged the median survival time of model mice, which might be due to the long circulation time, and enhanced accumulation of DOX in the glioblastoma tissues and pH-sensitive drug release. During treatment, the animal body weight of the saline group decreased sharply while that of the SL1-RNP-DOX group decreased slowly, indicating that SL1-RNP-DOX treatment can delay the effect of the progression of GBM (Figure 7B).

The histopathologic changes of GBM tissues reflected the therapeutic effect of different treatments intuitively through the TUNEL assay. As shown in Figure 7C, the SL1-RNP-DOX group exhibited the most cell apoptosis among all the treatment groups. The histological analysis was in keeping with the survival time of different DOX formulation groups. These results indicate that SL1 modification on RNP-DOX can enhance the therapeutic efficacy of RNP-DOX, which may be because SL1-RNP-DOX could accumulate more and penetrate more deeply into the glioblastoma tissues, subsequently producing more severe cytotoxicity than RNP-DOX. To preliminarily evaluate the safety of SL1-RNP-DOX in vivo, H&E staining of major organs after treatment was performed (Figure 7D), and it was found that almost no obvious necrosis or pathological changes occurred after SL1-RNP-DOX or RNP-DOX treatment.

## 4. Discussion

Polymer nanoparticles have been explored as promising drug delivery carriers due to their biodegradable and biocompatible properties [34,35]. However, as foreign materials, polymer nanoparticles can be recognized by the immune system after entering the blood circulation and cleared rapidly by the MPS. DOX is one of the most common and effective chemotherapeutic drugs, but it suffers from fast clearance and severe side effects. In this study, PLGA nanoparticles loaded with DOX were coated with SL1-modified RBC membranes to extend their half-life and enhance GBM-targeted delivery. The SL1-modified RBC membrane coating could significantly increase the size and zeta potential of NP-DOX, indicating the successful membrane coating on NP-DOX. The core-shell structure of SL1-NP-DOX was confirmed by TEM after uranyl acetate staining. The size of SL1-NP-DOX was around 80 nm which is reasonable for the nanoparticulate drug delivery system [36]. SL1-NP-DOX kept stable during storage in PBS due to that the membrane coating prevents the PBS-induced aggregation of NP-DOX. In vitro release revealed that SL1-RNP-DOX released DOX slowly under pH 7.4 but released DOX rapidly under pH 5.0, showing a superior pH-sensitive release. The results might be due to the unique structure of NP-DOX. DOX was loaded into PLGA nanoparticles via a non-surfactant emulsion method under a basic environment, where PLGA-COOH could be partly ironized (PLGA-COO-), become amphipathic under pH 8.0, and act as surfactants to help the formation of emulsions. NP-DOX could be stable under a physiological environment. However, in the acid environment (e.g., pH 5.0), amphipathic PLGA-COO- could be reprotonated, become lipophilic, and destabilize NP-DOX. Thus NP-DOX could dissemble to rapidly release DOX under pH 5.0 and demonstrate a pH-sensitive behavior, which is favorable for tumor chemotherapy [37]. 

The c-Met receptor plays an important role in the development and progression of tumors. There is increasing evidence that mutations and overexpression c-Met gene occur in various human cancers. Meanwhile, the high expression of c-Met is closely related to drug resistance and is one of the main reasons for the poor prognosis of cancer patients. Inhibitors of c-Met have shown promising potential in some clinical trials and have been popular molecular therapeutic targets for many years. Considering c-Met is overexpressed in U87MG cells, and the aptamer SL1 could bind to c-Met, we designed an SL1-modified RBC membrane-coated nanoparticle drug delivery system in this study. The aptamer SL1 was conjugated to DSPE-PEG to form aptamer SL1-modified lipid and inserted in RBC membranes. The surface density of ligands on nanoparticles is an important factor in determining the multivalent effect of ligand-modified nanoparticles, which can dramatically affect the targeting specificity of drug delivery systems. The surface SL1 density on NP-DOX was 110.5 SL1 molecules per nanoparticle, which is reasonable for a nano-size drug delivery system [31]. The interaction of SL1 and c-Met on the GBM cells would greatly increase the local concentration of nanoparticles on GBM cells and hence accelerate the receptor-mediated endocytosis. This was demonstrated by cellular uptake experiments where SL1-RNP uptake by U87MG cells was significantly improved compared with RNP. Moreover, the cell apoptosis assay demonstrated that SL1 modification on NP-DOX significantly enhanced the apoptosis of U87MG cells, probably due to more intracellular delivery of DOX through SL1-assisted active targeting.

The excellent GBM-targeting property of SL1-RNP in vivo was demonstrated through fluorescence imaging, biodistribution, and fluorescence slices in glioblastoma tissues. The antitumor effect of SL1-RNP-DOX was evaluated on orthotropic GBM models. It was found the antitumor effect of SL1-RNP-DOX was more powerful than RNP-DOX. These results not only confirmed that SL1-RNP could enhance drug delivery to brain tumors [38], but also suggested that SL1-RNP could further improve the antitumor effect of therapeutics.

## 5. Conclusions

In this study, we developed an active targeting pH-sensitive biomimetic nanoparticle (SL1-RNP-DOX) for c-Met-targeted therapy of GBM. SL1 modification on RNP facilitated drug delivery to GBM cells. In vitro studies demonstrated that U87MG cells had a higher uptake of SL1-RNP-DOX and a stronger pro-apoptosis effect than unmodified RNP-DOX. In vivo fluorescence imaging and tissue distribution further demonstrated higher tumor distribution of SL1-RNP-DOX compared with RNP-DOX. As a result, SL1-RNP-DOX presented the best anti-GBM effect with prolonged median survival time and the strongest tumor cell apoptosis in vivo among all groups. In conclusion, SL1-RNP-DOX exhibited a promising targeting delivery strategy for GBM therapy.

## Data Availability

Analyzed data are either presented in the article or can be provided upon request.

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
