# Peer review of "Aptamer-Modified Erythrocyte Membrane-Coated pH-Sensitive Nanoparticles for c-Met-Targeted Therapy of Glioblastoma Multiforme"

_membranes, 2022, doi:10.3390/membranes12080744_

Round 1

Reviewer 1 Report

This manuscript describes the development of an active targeting pH-sensitive biomimetic nanoparticle. The manuscript is interesting and to be overall well written. Not found any critical comments.

I think this manuscript is acceptable for publication.

Author Response

Dear Reviewer,

Many thanks for your encouragement and best wishes to you!

Xianping Liu

2022.07.10

Reviewer 2 Report

It is interesting that SL1-RNP-DOX exhibited a promising targeting delivery strategy for GBM therapy. How about the yield of RBC membranes? Even authors mentioned that RNP, usually consisting of a biodegradable and biocompatible poly(lactic-co-glycolic acid)(PLGA) nanoparticle core and an RBC membrane shell, represents a promising brain drug delivery system due to their combination of high drug carrying capacity along with an inherently biocompatible membrane coating, there is no comparison data without RBC membrane.

Author Response

Dear Reviewer,

Thanks for your review of the manuscript. We have carefully revised the manuscript according to your suggestion. Point-by-point responses to your comments are listed as follows:

Question 1: It is interesting that SL1-RNP-DOX exhibited a promising targeting delivery strategy for GBM therapy. How about the yield of RBC membranes?

Answer: Thanks for the question. The preparation procedure of RBC membranes is provided in Section 2.4. Generally, 1 ml of RBC membranes containing around 2.0 mg of membrane proteins could be generated from 1 ml of the whole blood. The result agrees well with our previous report (ACS Nano 2019, 13:4148-4159). This result has been added to the revised manuscript.

Question2: Even authors mentioned that RNP, usually consisting of a biodegradable and biocompatible poly(lactic-co-glycolic acid)(PLGA) nanoparticle core and an RBC membrane shell, represents a promising brain drug delivery system due to their combination of high drug carrying capacity along with an inherently biocompatible membrane coating, there is no comparison data without RBC membrane.

Answer: Thanks for the comments. In this manuscript, we compare the physio-chemical property of SL1-RNP-DOX with RNP-DOX and NP-DOX without RBC membrane coating. Considering that NP-DOX without RBC membrane coating is not stable in PBS (Fig. 1E) and the goal of this study is to develop an active targeting pH-sensitive RNP loaded with DOX (SL1-RNP-DOX) for c-Met-targeted therapy of glioblastoma multiforme, RNP-DOX rather than NP-DOX is used as a control in the following experiments. We will carefully consider your questions in our further study.

Best regards

Author: Xianping Liu

2022.07.10

Reviewer 3 Report

It was a nice study about the fabrication and evaluation of the aptamer functionalized pH-responsive RNP nanoparticles for GBM treatment application. Here are some comments on this study that should be considered before publication:

1-      There are some grammatical mistakes in the text that should be corrected.

2-      Please add some of the statistical results in the abstract.

3-      Why did you choose 1 week for stability evaluation? Moreover, it is better to evaluate changes in the amounts of loading and release pattern of formulation during the storage time.

4-      Why did you use coumarin-6 loaded carrier for cellular uptake test, while DOX itself has fluorescence property?

5-      For better comparison, please redraw the cell viability curve of DOX-contained carriers based on the concentration, not logarithmic form.

6-      Please describe more about the results of the apoptotic assay.

7-      Please add the results of the drug-loaded cytotoxicity test against a type of normal cell line too.

8-      Please add the results of cell uptake of nanoformulation during times (after 6, 12, and 24h).

9-      According to the biodistribution results, most of the nanoformulations are accumulated in other organs than the brain, even in the case of the targeted sample. How do you explain this? Do you think it is good?

10-   Please add the results of H&E staining of GBM treated with drug-loaded samples.

11-   Please compare the results of your study with other similar research in the discussion part.

12-   Please use more updated references. 

Author Response

Dear Reviewer,

Thanks for your review of the manuscript. We have carefully revised the manuscript according to your suggestion. Point-by-point responses to your comments are attached.

Thanks again to the reviewer for your hard work! Best wishes to you!

Author: Xianping Liu

2022.07.10

Round 2

Reviewer 2 Report

The manuscript was well revised and I recommend to accept in present form.

Reviewer 3 Report

Thanks for addressing the comments.